# Cross-Sectional Study to Evaluate Knowledge on Hand Hygiene in a Pandemic Context with SARS-CoV-2

**DOI:** 10.3390/medicina58020304

**Published:** 2022-02-17

**Authors:** Cătălina Iulia Săveanu, Andreea Porsega, Daniela Anistoroaei, Cristina Iordache, Livia Bobu, Alexandra Ecaterina Săveanu

**Affiliations:** 1Department I—Surgicals, Faculty of Dental Medicine, University of Medicine and Pharmacy Grigore T Popa, 700115 Iasi, Romania; cisaveanu@prevod.umfiasi.ro; 2Faculty of Dental Medicine, University of Medicine and Pharmacy Grigore T Popa, 700115 Iasi, Romania; andreeapor@yahoo.com (A.P.); alexandrasaveanu@yahoo.com (A.E.S.); 3Department of Implantology, Removable Prostheses, Dental Technology, Faculty of Dental Medicine, University of Medicine and Pharmacy, 700115 Iasi, Romania

**Keywords:** hand hygiene, antiseptic solution, decontamination

## Abstract

*Background and Objectives*: The basis of any infection control program is hand hygiene (HH). The aim of this study was to investigate knowledge of HH among medical students. *Materials and Methods*: Students were randomly selected from two Romanian universities and a cross-sectional, questionnaire-based study was conducted between January and May 2021. The answers regarding demographic data and knowledge concerning the methods, the time and the antiseptics used for HH were collected. The selection of the study group was made according to selection criteria in accordance with ethical issues. A descriptive statistical analysis was performed, and a chi-square test was used for data comparison, with a cut-off point of 0.05 for statistical significance. *Results*: The results indicated that the attitude of the students towards the practice of HH improved significantly. Most students believe that simple HH can control infections. Significant differences were found by the year of study in terms of the hand surfaces included and recommended duration (*p* < 0.05). *Conclusions*: In conclusion, the study shows that most respondents have sufficient knowledge on HH, meaning that a higher compliance is required to control infections. The indicated reasons of non-compliance with HH are emergencies and other priorities.

## 1. Introduction

Infection control in dentistry is a broad concept on several levels. The increase in the incidence of certain diseases, including SARS-CoV-2 infection, is also linked to the lack of compliance of medical staff on various issues. Hand hygiene (HH) is the basis of any program for infection control [1,2,3,4,5,6]. However, the promotion of hand hygiene behavior remains a complex issue [3,7,8]. Task forces for discussion and expert consensus on critical issues related to hand hygiene in health care are based on behavioral changes, education, training, tools, World Health Organization (WHO)-recommended hand antisepsis formulations, glove use, water quality for handwashing, patient involvement, religious and cultural aspects of hand hygiene, indicators for service implementation and monitoring, regulation and accreditation, communication/campaigning, and national guidelines on hand hygiene [9]. Hand hygiene is a good predictor of nosocomial infections, especially when the medical staff come in contact with a patient’s oral cavity [10]. HH is a widespread topic among students or practitioners. Aspects regarding HH have been addressed in numerous studies conducted during the pandemic period with SARS-CoV-2 [11,12,13].

Wearing gloves does not provide absolute protection and, therefore, HH takes on extra value. The dynamics of hand contamination are similar on gloved versus ungloved hands; gloves reduce hand contamination, but do not fully protect from the acquisition of bacteria during patient care. Therefore, the glove surface is contaminated, making possible cross-transmission through contaminated gloved hands [9]. The WHO stated “The five moments for hand hygiene”, namely: before contact with the patient, before performing any procedure, after exposure to biological fluids at risk, after contact with the patient, and after contact with objects around the patient [9].

The methods of HH are: simple washing with soap and water, washing with water and antiseptic soap, rubbing the hands with antiseptic solution (60–95% alcohol) and surgical washing.

HH with soap and water promotes the mechanical removal of organic matter, visible dirt and microorganisms from the skin surface [9]. Transient flora (transient microbiota), which colonizes the superficial layers of the skin, is more amenable to removal by routine HH. Transient microorganisms do not usually multiply on the skin, but they survive and sporadically multiply on the skin surface [9,14,15]. Normal human skin is colonized by bacteria, with total aerobic bacterial counts ranging from more than 1 to 1 × 10 CFU/cm^2^ on the forearm [9,16].

HH is considered by some researchers to be more effective than decontamination with antiseptic solutions, as it reduces the microbial load by eliminating transient flora and organic matter [17]. Other studies have shown that washing with soap and water for 30 s leads to the elimination of noroviruses from the hands [18] and an increase of the washing time decreases the number of microorganisms on hand surfaces [19].

HH with antiseptic solution is increasingly used, due to its easy and fast application, and proven efficiency; it reduces transient flora and resident flora, but not organic matter [9]. WHO recommends alcohol-based hand rubs based on the following: evidence-based, fast-acting and broad-spectrum microbicidal activity with a minimal risk of generating resistance to antimicrobial agents; suitability for use in resource-limited or remote areas with lack of accessibility to sinks or other facilities for hand hygiene; and capacity to promote improved compliance with hand hygiene by making the process faster and more convenient [9]. The effectiveness of HH with antiseptic solution depends on the type of alcohol (the most commonly used are chlorhexidine gluconate, iodophors and triclosan), the amount applied (usually 3 mL) and the technique used [20]. The Center for Disease Control and Prevention (CDC) published in 2020 the Guideline for HH in HealthCare Settings, an update of the guide developed in 2002 [2]. According to the Guideline, in all the cases of contact with the patient, HH with antiseptic solution is recommended, instead of the traditional method of simple HH (unless the hands are visibly dirty) [21].

A lot of information is available regarding HH. However, it is not very well known, or not properly applied, and this is why the pathogenic microbial flora of the hands is still involved in the transmission of diseases. Numerous studies have highlighted gaps in hand hygiene behaviors [22,23,24]. Because dental clinical activity involves the direct contact of medical staff with biological fluids, HH is very important and must be fully known by all future practitioners. In this context, university education aims to build the professional behavior of future specialists.

The aim of the present study was to investigate knowledge about HH practices among students attending two medical universities in Romania.

The null hypothesis was that there are no differences in knowledge about HH, regardless of the educational institution and the year of study.

The testable hypothesis was that there are differences in knowledge about HH, depending on the educational institution and the year of study. This research was used to identify gaps in knowledge, attitude toward and practices of hand hygiene among university students. The results of this study are very useful to motivate the implementation of behavior change programs on measures to induce proper hand hygiene.

## 2. Materials and Methods

The questionnaire method was used for the assessment of students’ level of knowledge.

### 2.1. The Survey

A cross-sectional, questionnaire-based study was conducted between January and May 2021. The 51-item questionnaire used was evaluated by a panel of experts from the Faculty of Dentistry within the “Grigore T. Popa” University of Medicine and Pharmacy in Iasi, Romania, following a qualitative pre-testing of the content and validation. The questionnaire was openly applied and it was uploaded online on the Google docs platform. The 51 items referred to: demographic data (Q1–Q4); data on the frequency of HH in relation to the patient and other situations, and on the use of HH products (Q5–Q15); data regarding infection control by simple hand decontamination, and the substances used (Q16–Q34); data regarding infection control by surgical decontamination (Q35–Q42); and data regarding the antiseptic substances for HH (Q43–Q51).

### 2.2. The Study Group

The study included students randomly selected from two universities: “Grigore T. Popa” University of Medicine and Pharmacy in Iasi (UMP1) and “George Emil Palade” University of Medicine, Pharmacy, Science and Technology in Targu Mures (UMP 2) Table 1. The selection of the study group was made following selection criteria in accordance with ethical issues and good practices of study. The inclusion criteria were: students enrolled in a form of education with a medical profile; students who consented to participate in the study; students who agreed to fill in the questionnaire; students attending years 3 to 6. The exclusion criteria were: students who did not attend a university with a medical profile; students in the first and second years of study, as they did not go through the preparation module regarding infection control. Sampling of the respondents was unlikely. The students considered eligible were those who agreed to complete the questionnaire after reading its content. A total of 126 subjects completed the questionnaire.

### 2.3. Data Collection

For the data collection, the following closed-ended or variable-answer questions were applied as follows: The data were collected and introduced into a database. SPSS 26.00 for Windows (IBM, Armonk, NY, USA) was used for the processing of statistical data. A descriptive statistic of the study was performed by applying crosstabs to all the aspects analyzed according to MS and HH. The Chi-square test was used for data comparison. The cut-off point of statistical significance *p* was set at 0.05.

## 3. Results

The mean age of the students was 23.15 years (± 2.083), with the youngest being 19 years old and the eldest being 37 years old. The general characteristics of the study group are presented in (Table 1).

Attitude regarding HH before and during the SARS-CoV-2 pandemic indicated that HH was performed by 42.9% of the students more than 10 times a day during the pandemic, compared to 10% before its onset. Statistically significant differences were found only by the year of study (*p* < 0.05) (Table 2), with students in the 6th year performing HH more frequently.

No statistically significant differences were found in terms of educational background or gender. Most students always perform HH before and after contact with the patient, before applying gloves, after removing gloves, after using the toilet, before meals, and when hands are visibly dirty (Table 2).

Statistically significant differences were found in hands washing after using the toilet by year of study, in favor of sixth-year students (*p* < 0.05) and in hand washing when hands are visibly dirty by the university attended (*p* < 0.05), with a higher percentage for students attending the university in Iasi (61.9%) compared to those attending the UMP2 (30.95%). Female students use antiseptic soap more often than male students (*p* < 0.05). Most students consider that simple hand decontamination can prevent the transmission of infections after contact with the patient, followed in descending order by: immediately after exposure to biological fluids (saliva, blood), after touching the areas near the patient, and before aseptic procedures. Although differences were found in the response rate percentage regarding infection control by hand decontamination and the frequency of solutions used, these differences were not statistically significant by gender, background, specialization, university or year of study (*p* ˃ 0.05). A percentage of 15.9% (20) of the investigated students do not know that washing with soap and water removes organic matter and transient flora from the skin surface, and 43.7% of them (55) consider that antiseptic soap is more effective than antiseptic solutions.

The vast majority of students agree that antiseptic solutions are used when soap and water are not available and 14.3% (18) do not know that rubbing hands with antiseptic solutions reduces the resident flora on the skin surface. A considerable percentage of the students, 77% (97) use antiseptic solutions when the hands are not visibly dirty (Table 3).

A percentage of 40.5% (51) of the investigated students state that they have “to a very large extent”sufficient knowledge, whereas 50.8% (64) of them consider that they have “largely” sufficient knowledge regarding the techniques of hand decontamination. Most students claim that they know that hand decontamination is a measure to control infection. Only 48.4% of them (60) feel guilty and 44.4% (56) feel frustrated when they fail to perform proper hand hygiene. Not all students follow the recommendations of the World Health Organization guide on HH. No significant differences were observed by gender, background, specialization, university or year of study (*p* ˃ 0.05) (Figure 1).

Regarding surgical hand washing, almost all students consider the following issues to be important: removal of jewelry, watch and artificial nails, as well as shortening of nails before surgical HH. A percentage of 70.6% (89) consider that antiseptic soap with 4% chlorhexidine (CHX) can be used for surgical washing, and 73.0% (92) know that the hands should be kept up and the elbows should be kept down during surgical washing. A large percentage of students, 96.8% (122), know the surfaces of the hands that need to be decontaminated during surgical washing. Significant differences were found by year of study regarding the hand areas for decontamination, by year of study, specialization and gender regarding the positioning of hands and elbows, and by gender regarding forearm inclusion in surgical HH (*p* < 0.05) (Table 4).

Most of the students consider that the time required to wash their hands with soap and water is of 30–60 s, and the time required to decontaminate their hands with antiseptic is 20 s (Table 5). The duration of surgical HH was estimated at 2–3 min or 5 min by approximately equal percentages of students. The concentration of ethanol in antiseptic was estimated at 70% by half of the investigated students, and the required amount of antiseptic solution to be used was estimated at 3–5 mL by 69.8% (88) of the students. A little more than a half of respondents consider that the residual action of antiseptics is due to CHX. For hand drying, 52.4% (66) of students use a cotton towel and 45.2% (57) use paper towels. The multitude of answers regarding the importance of washing hands with soap and water confirms the hypothesis. Faculty courses were the main source of information on HH for 65.9% (83) of the students. Significant differences (*p* < 0.05) were found in the recommended duration for HH by year of study, in the usual concentration of ethanol in antiseptic, in the reason for a residual action and in the method of hand drying by specialization, and in hand washing with soap and water as a habit by gender.

## 4. Discussion

Knowledge of HH infection control methods is essential for students [3,10,11,12,13,25,26]. HH is a routine procedure, which aims to remove organic matter and transient flora. The use of an antiseptic soap will determine, in addition to the cleansing action by the physical removal of organic matter, the inactivation of microorganisms, by the action of antiseptic substances in its composition [3]. The results of the present study show that the level of students’ knowledge on the five moments of HH is high. In this study, we did not assess the level of knowledge before the onset of the SARS-CoV-2 pandemic, but the results of other studies indicate that the majority of respondents (83%) had a high knowledge of HH, due to the development of student training programs [27]. However, another study conducted before the onset of the SARS-CoV-2 pandemic found that only 63% of medical students included in their study knew the five moments of HH [28].

Another parameter considered was the duration of simple and antiseptic decontamination of the hands. Approximatively half of all respondents (54%) know the recommended duration of 20–30 s for simple HH whereas 38% consider that 20 s is enough for complete hand cleaning. In the field of HH, there is a European standard for the evaluation of the effectiveness of antiseptic agents for HD: EN 1499 (HH washing), EN1500 (HH rubbing) and EN12791 (surgical HH) [10]. However, many students enrolled in the present study consider that glycerin is used for its antiseptic action and, moreover, that it would have a residual action. The fact that some students consider that triclosan has remanent action and not CHX again indicates a low level of knowledge. Compared to simple soap, the one containing 0.3% triclosan did not lead to a significant reduction in microorganisms, except *E. faecalis* [10,29]. In addition, the use of triclosan is controversial because of several side effects that have been reported, such as carcinogenic effects, allergies, endocrine disorders, acute or chronic toxicity, antibiotic resistance. Therefore, in 2016, the US Food and Drug Administration banned triclosan and 17 other ingredients in antiseptic soaps, requesting more data on the efficacy and safety of these antiseptic agents [30].

A particularly important aspect is the fact that 44.7% (67) of the students do not know the action of CHX although, currently, it is considered to be the most frequently used antiseptic agent. In this context we can say that the ignorance of the advantages of using CHX implicitly attracts its non-use in practice. CHX is used in concentrations of 2% and 4% with a very wide spectrum of action and a residual action that lasts several hours [31]. CHX in the 0.5% concentration may increase the effectiveness of alcohol-based antiseptic. CHX is inactivated by anionic agents, and for this reason, it is recommended to avoid the use of creams or soaps [32,33].

The use of iodophors was indicated by only 10.7% (16) of the students. The majority of students did not know that the iodophors are antibacterial agents with a broad spectrum of activity and a prolonged action.

Alcohol is the most effective, the safest and the most widely used HH agent. Ethyl alcohol has a broad spectrum of activity at an increased concentration of 70–80%, but with limited action in eliminating spores [10]. The effectiveness of alcohol is influenced by many factors such as: the type of alcohol used and its concentration, the contact time, the amount applied and skin moisture. In vitro studies have shown that hydroalcoholic solutions containing 60–80% alcohol have reduced bacteria by 4-log to 6-log in 30 s; a shorter application time decreases the efficiency of the hydroalcoholic solution and is associated with the use of a smaller amount of product [34]. N-propanol is effective against hepatitis B virus, HIV, influenza A virus, rotavirus, adenovirus and bacteria at 60–90% concentration. HH with alcohol-based agents is the key measure in the prevention of healthcare-associated infections and nosocomial transmission of pathogens. A study conducted in Germany which involved dental students showed that improvements in general knowledge and special efforts were needed to increase compliance with HH [35].

The time required to rub hands with hydro-alcoholic solutions was known by 65.3% of students involved in this study. This result is in contrast to a study conducted in Pakistan, which reported that only 20% of respondents were aware of this issue [36]. A prospective study in an intensive-care and pediatrics unit in France showed that only a third of students knew the appropriate duration of hand rubbing before theoretical training and, furthermore, the duration of hand decontamination was found to decrease significantly during repeated procedures [37].

It was noteworthy that the stages of preoperative preparation of the hands were known by more than 90% of the students, and almost all respondents (97.3%) stated that the areas that need to be decontaminated are the palmar and dorsal face of the hands, the interdigital area, the thumbs, the fingertips and the wrist. On the other hand, the answers regarding surgical HH technique showed that the respondents have knowledge regarding surface area: although 97.3% of them stated that surgical HH includes the forearms, only one third (31.3%) had deeper knowledge of the notion of the correct position of the forearms during the preoperative preparation of the hands. HH is a behavioral practice, so it is important to identify the reasons for non-compliance with recommendations. In the current study, 6.7% of students answered that emergencies and other priorities make it difficult to practice HH; however, 98.7% noted that they feel guilty when they fail to perform HH, and 98% feel frustrated when those around them omit HH. Similar findings were found in a study conducted in India on a group of 130 students, where about 40% of medical students said that emergencies cause them to practice HH with difficulty and that they considered they could adhere to good HH practices if hygiene facilities were adequate and handy [38].

In this study, it was observed that students wash their hands more than 10 times a day after the onset of the SARS-CoV-2 pandemic. A similar result was reported by Dwipayanti et al. in Indonesia [12].

Hand drying is a routine procedure but certain aspects must be taken into account, namely: the cotton towel can be recommended but should be used only once and then reprocessed because the humid environment will favor microbial colonization; the disposable paper towel is the best option—in addition to drying, it also removes the bacteria from the skin. A hot air dryer is not recommended because it causes the aerial dissemination of microorganisms, especially when the hands are rubbed together, and, in addition, requires a longer use time of about 40 s, produces noise, causes excessive drying of the skin and should be cleaned often [39]. Factors associated with the choice of hand-drying method include the availability of the method, the possibility of minimizing contact with the surrounding surfaces, the perception and the rapidity of the method [40]. On the other hand, there are studies showing that, instead of electric dryers, the use of paper towels reduces the spread of bacteria, recommended in hospitals or in areas at high risk of cross-contamination, [41]. In recent years, a new version of the hot air dryer has been introduced, the air-jet dryer with antibacterial filter. It has a similar efficiency to disposable wipes and requires a relatively shorter drying time compared to the hot air dryer, which helps to improve compliance. However, it has been noticed that the dispersion of bacteria in the air can reach up to 3 m, due to the high speed of air emission during use, therefore such devices are not recommended within medical facilities [42,43].

The results of the present study show that medical and teaching staff can provide useful and complex information about hand hygiene through lectures, and they can be models for students, leading them to properly practice HH techniques. In addition, prevention lectures are considered to be a reliable source of information by the investigated students, a fact shown by the results of similar studies [11,12,13,14,28,36,38]. The results of the present study show that medical and teaching staff can present useful and complex information about hand hygiene in the courses and can be models for students, leading them to properly practice HD techniques. In addition, prevention lectures are considered a reliable source of information by the investigated subjects. The media also play an important role through public information programs. This study has some limitations that need to be considered: the small number of participants in this study, the distribution by gender, year of study or specialization was uneven, the subjects were randomly selected and the bias of any analyzed group was not followed.

## 5. Conclusions

Within the limits of this study, we can admit that educational measures are needed in order to guide medical staff for 100% compliance with the knowledge and adoption of HH measures. In order to prevent the transmission of microorganisms between patients and dental staff, high compliance with infection control practices is required. The study shows that most students know the moments of HH, meaning that high compliance is needed to prevent infection control. The causes of non-compliance with hand decontamination rules are emergencies and other priorities. Most students explain, to some extent, to those around them the appropriate HH technique, an attitude of responsibility emphasized also by the fact that most students follow WHO recommendations. The most commonly used product for drying hands was the cotton towel (51.3%), followed by the paper towel (44%) and the hand dryer (2.7%). Only 2% said they do not dry their hands. Unfortunately, half of the students use a cotton towel to dry their hands instead of a paper towel. A very small percentage of students are unaware of the disadvantages of drying their hands with a hand dryer and, therefore, use it. The most important sources of information on the correct HH are faculty courses and the media. The study highlights the need to improve training programs to achieve a higher level of knowledge related to hand hygiene.

## Figures and Tables

**Figure 1 medicina-58-00304-f001:**
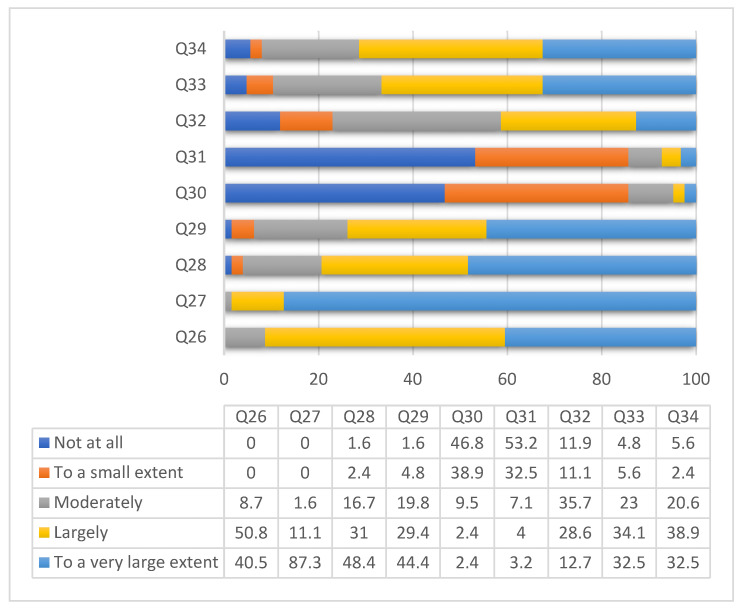
Distribution of answers concerning knowledge on hand decontamination. The questions were as follows: Q26 = I have sufficient knowledge of HH techniques; Q27 = I consider HH to be a measure of infection control; Q28 = When I fail to do proper HH I feel guilty; Q29 = When others omit HH, I feel frustrated; Q30 = Sometimes I forget to make HH; Q31 = Emergencies make me not practice HH; Q32 = Sometimes I explain to others the correct HH technique; Q33 = I use the HH technique recommended by WHO; Q34 = I perform HH with antiseptic according to WHO recommendations.

**Table 1 medicina-58-00304-t001:** General characteristics of the study group.

Q			Year of Study				Total	
			3rd	4th	5th	6th		
			*N*	*N*	*N*	*N*	*N*	%
		Total	29	31	18	48	126	100
Q1	Gender	Male	6	9	7	10	32	25.4
		Female	23	22	11	38	94	74.6
Q2	Area	Rural	13	10	11	15	49	38.9
		Urban	16	21	7	33	77	61.1
Q3	University	UMP1	26	3	10	43	82	65.1
		UMP2	3	28	8	5	44	34.9
Q4	Specialization	Dental Medicine	26	7	8	26	67	53.2
		General Medicine	3	24	10	22	59	46.8

**Table 2 medicina-58-00304-t002:** Frequency of answers to questions concerning hand hygiene (HH) in relation to the patient and other situations and the use of HH products. Results of statistical significance tests comparing the frequency of HH in relation to the patient and the use of HH products by variables.

Questions		Frequency of Responses %(Count)					G	A	U	S	Y
							*p*	*p*	*p*	*p*	*p*
		0/Day	1–2/Day	3–5/Day	6–10/Day	>10/Day					
Q 5	Frequency of simple hand washing before the SARS-CoV-2 pandemic	0.8(1)	4.8(6)	43.7(55)	39.7(50)	11.1(14)	0.376	0.591	0.09	0.32	0.040
Q 6	Frequency of simple hand washing during the SARS-CoV-2pandemic	0	1.6(2)	17.5 (22)	38.1(48)	42.9(54)	0.504	0.715	0.66	0.07	0.42
		Never	Rarely	Sometimes	Often	Always					
Q 7	Do you perform HH before and after contact with the patient?	0	0	3.2(4)	0.8(1)	96(121)	0.84	0.14	0.25	0.38	0.37
Q 8	Do you perform HH before applying gloves?	0.8(1)	0	7.3(11)	9.3(14)	82(123)	0.44	0.41	0.20	0.01	0.14
Q 9	Do you perform HH after removing gloves?	0	0	4(5)	15.9(20)	80.2(101)	0.09	0.60	0.20	0.92	0.17
Q 10	Do you perform HH after using the toilet?	0	0	0.7(1)	0	96.7(145)	0.85	0.13	0.14	0.37	0.38
Q 11	Do you perform HH before meal?	0	0	3.2(4)	2.4(3)	94.4(119)	0.95	0.19	0.14	0.59	0.04
Q 12	Do you perform HH when hands are visibly dirty?	0.8(1)	0	3.2(4)	3.2(4)	92.9(117)	0.95	0.24	0.01	0.09	0.44
Q 13	How often do you use soap?	1.6(2)	2.4(3)	9.5(12)	9.5(12)	77(97)	0.61	0.83	0.08	0.66	0.93
Q 14	How often do you use antiseptique soap?	7.9(12)	20.6(26)	30.2(38)	24.6(31)	16.7(21)	0.02	0.23	0.59	0.70	0.92
Q 15	How often do you use antiseptique gel?	0	6.3(8)	17.5(22)	36.5(46)	39.7(50)	0.10	0.26	0.07	0.29	0.06

G = gender, A = area, U = University, S = specialization, Y = year of study, *p* = significance level.

**Table 3 medicina-58-00304-t003:** Response rate percentage regarding infection control by simple hand decontamination. Results of statistical significance tests.

Question		Frequency		G	A	U	S	Y
		Yes	No	*p*	*p*	*p*	*p*	*P*
		%(Count)	%(Count)					
Q16	Antiseptic solutions are used when hands are not visibly dirty?	77(97)	23(29)	0.51	0.58	0.09	0.06	0.05
Q17	Antiseptic solutions reduce the resident flora from the skin surface?	85.7(108)	14.3(18)	0.13	0.12	0.70	0.42	0.73
Q18	Antiseptic solutions are used when soap and water are not available?	82.5(104)	17.5(22)	0.16	0.79	0.25	0.54	0.21
Q19	Antiseptic soap is more effective than antiseptic solutions?	43.7(55)	56.3(71)	0.40	0.38	0.11	0.93	0.01
Q20	Does soap remove organic matter and transient flora?	84.1(106)	15.9(20)	0.24	0.54	0.30	0.08	0.57
Q21	Can simple hand decontamination control the infections before aseptic procedures?	84.9(107)	15.1(19)	0.64	0.84	0.85	0.29	0.60
Q22	Can simple hand decontamination control the infections after touching the surfaces near the patient?	85.7(108)	14.3(18)	0.40	1.0	0.88	0.83	0.90
Q23	Can simple hand decontamination control the infections after exposure to biological fluids?	85.7(108)	14.3(18)	0.74	0.29	0.88	0.42	0.28
Q24	Can simple hand decontamination control the infections after direct contact with the patient?	93.7(118)	6.3(8)	0.09	0.11	0.09	0.10	0.80
Q25	Can simple hand decontamination control the infections before direct contact with the patient?	91.3(115)	8.7(11)	0.88	0.86	0.15	0.24	0.42

G = gender, A = area, U = University, S = specialization, Y = year of study, *p* = significance level.

**Table 4 medicina-58-00304-t004:** Response rate percentage regarding infection control by surgical decontamination. Results of statistical significance tests.

Question		Frequency		G	A	U	S	Y
		Yes	No	*p*	*p*	*p*	*p*	*p*
		%(Count)	%(Count)					
Q35	Do you consider it important to remove jewelry prior to surgical HH?	97.6(123)	2.4(3)	0.31	0.84	0.19	0.49	0.17
Q36	Do you consider it important to remove watch prior to surgical HH?	95.2(120)	4.8(6)	0.65	0.15	0.93	0.39	0.45
Q37	Do you consider it important to shorten nails prior to surgical HH?	97.6(123)	2.4(3)	0.75	0.84	0.24	0.06	0.34
Q38	Do you consider it important to remove artificial nails prior to surgical HH?	99.2(125)	0.8(1)	0.56	0.42	0.46	0.29	0.65
Q39	Do you wash your hands with 4% CHX soap?	70.6(89)	29.4(37)	0.13	0.58	0.39	0.90	0.35
Q40	The areas to be decontaminated are: palmar and dorsal face, interdigital area, thumbs, fingertips and wrist?	96.8(122)	3.2(4)	0.99	0.56	0.52	0.25	0.004
Q41	D you keep your hands up and elbows down during surgical HH?	73(92)	27(34)	0.01	0.46	0.23	0.0	0.006
Q42	Does surgical HH include the forearms?	96.8(122)	3.2(4)	0.02	0.64	0.14	0.06	0.68

G = gender, A = area, U = University, S = specialization, Y = year of study, *p* = significance level.

**Table 5 medicina-58-00304-t005:** Frequency of answers to questions concerning antiseptic substances for HH. Results of statistical significance tests comparing the frequency of answers on antiseptic substances for HH, by variables.

Question		Interval					G	A	U	S	Y
		Frequency %(Count)					*p*	*p*	*p*	*p*	*p*
Q43	The duration of HH with soap	10 s	20 s	30–60 s	2 m		0.23	0.82	0.17	0.11	0.02
		0.8(1)	38.1(48)	54(68)	7.1(9)						
Q44	The duration of HH with antiseptic solution	10 s	20 s	30–60 s	2 m		0.62	0.87	0.28	0.83	0.05
		18.3(23)	63.5(80)	18.3(23)	0.0(0)						
Q45	The duration of surgical HH	1–2 m	2–3 m	5 m	10 m		0.57	0.41	0.73	0.05	0.61
		0.0(0)	42.9(53)	43.7(54)	13.5(17)						
Q46	What is the concentration of ethanol in antiseptic solutions?	40–60%	70%	90%	I don’t know		0.86	0.06	0.22	0.03	0.09
		18.3(23)	54.8(69)	5.6(7)	21.4(27)						
Q47	What is the recommended amount of antiseptic?	1–2 mL	3–5 mL	7 mL	10 ml		0.94	0.35	0.87	0.89	0.74
		22.2(28)	69.8(88)	7.9(10)	0.0(0)						
Q48	Antiseptics have a residual action if they contain:	Glycerin	CHX	Triclosan	Iodoform		0.05	0.13	0.39	0.05	0.53
		19.8(25)	54(68)	14.3(18)	11.9(15)						
Q49	What do you use most often to dry hands?	Cotton	Paper	Gown sleeve	Dryer	I don’t dry	0.79	0.59	0.16	0.00	0.90
		52.4(66)	45.2(57)	0.8(1)	1.6(2)	0.0(0)					
Q50	Has washing your hands with soap become a habit?	Not at all	To a small extent	Moderately	Largely	Very largely	0.01	0.25	0.32	0.31	0.21
		0.0(0)	0.0(0)	0.8(1)	7.1(9)	92.1(116)					
Q51	What were your sources of information?	Media	Faculty	Family	Friends		0.16	0.92	0.86	0.76	0.43
		17.5(22)	65.9(83)	10.3(13)	4.8(6)	1.6(2)					

G = gender, A = area, U = University, S = specialization, Y = year of study, *p* = significance level, s = secondes; m = minutes, CHX= Chlorhexidine

## Data Availability

The data that support the findings of this study are available on request from the corresponding author.

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
