# Peer review of "Cross-Sectional Study to Evaluate Knowledge on Hand Hygiene in a Pandemic Context with SARS-CoV-2"

_medicina, 2022, doi:10.3390/medicina58020304_

Round 1

Reviewer 1 Report

An interesting hand washing study. A study that is much needed, especially disseminated to medical and dental schools. The practice of handwashing challenges could be described further (are we talking about in med/dental school or when they are in the hospital?). Overall good job. Several recommendations to help get your papers publish is below:

The tables will need to be fixed. See my comments in the manuscript.

There are too many acronyms that need to be spelled out for clarity.

The authors begin with percentages in their sentences. Consider writing the percentage numbers or begin with a vowel.

The authors need to choose one identification of participants, subjects, or respondents. 

Discussion section need to address adding citations, especially when they write "studies," and there is no citations behind their sentence. And the follow-up next sentence has one study described. 

The authors need to focus a little more on attitudes in their results and discussion. That part is weak (or remove "attitudes"). If an attitudinal survey was used, how was the survey developed? Was it by an existing survey or developed one based on their experience? Regarding the hand washing survey - how was the survey developed? Was it pilot-tested?

Author Response

Thank you very much for your pertinent and valuable recommendations.

Considering the suggestions received from the reviewers, the following changes were made to the article.

Corrections were made to the Abstract section, as well as to the Introduction section.

Subsequent paragraphs of the Material and Methods section were numbered.

Students from only 2 universities were chosen because that's how we succeeded in communication. Initially, the questionnaire was sent to a higher number of students.

 The first- and second-year students were eliminated considering that they did not go through the preparation module regarding infection control, which could have led to processing errors.

To avoid other misunderstandings, we rewrote the protocol, taking into account the 126 students.

The questionnaire was evaluated by a panel of experts from the Faculty of Dentistry, following a qualitative pre-testing of the content, followed by its validation. The questionnaire was openly applied. The questionnaire was uploaded online on the Google docs platform.

 All the information obtained was collected after the pandemic, with the establishment of the level of knowledge at that level.

I improved the clarity of the tables.

I gave up some acronyms.

I corrected the sentences that started with a percentage.

I renamed the subjects as students.

I made the corrections for Discussions.

I removed attitudes.

I am attaching the corrected form of the article.

Thank you!

Best regards,

Iulia Saveanu

Reviewer 2 Report

General: The manuscript of Săveanu et al. presents an interesting and up-to-date topic of hand hygiene practices among students of medical universities. However, there are some flaws which should be corrected and addressed. Unfortunately the Materials and Methods sections is poorly written and many details about the recruitment  are lacking. Authors should pay attention to text formatting, as there are double spaces throughout the manuscript. I would also suggest to have the manuscript proofread by the professional English agency as some sentences are hard to follow. I have listed the detailed remarks below:

Abstract

  • There is no need to write the names of subsequent parts of the abstract, like background and objectives, materials and methods and so on. I suggest removing it.
  • Line 39 – Romanian, not romanian. Please also add what was the size of the sample.
  • I suppose the study was conducted in the period of the COVID-19 pandemic, so the exact period of the study should be added.
  • Line 33 – according to
  • Line 36 – if Authors use an abbreviation for the first time, it should be explained.
  • Line 38 – what is D? It is unclear.

Introduction

  • I have on major objection regarding this introduction. As your study was conducted in May 2021 and focuses on evaluation of hand hygiene behaviors among students, it would be much better to present some studies which were also conducted in the period of COVID and also addressed the matter of hand hygiene. Now there is unclear whether articles which you cite were carried out in the COVID period. Thera are some articles which may be helpful:

https://www.mdpi.com/1660-4601/18/2/409

https://www.ncbi.nlm.nih.gov/pmc/articles/PMC8186824/

https://www.ncbi.nlm.nih.gov/pmc/articles/PMC8373300/

Material and Methods

  • Subsequent paragraphs should be numbered (for example 2.1. Study Sample, 2.2. Applied Questionnaire etc.)
  • There is no information whether Authors obtained the consent of Institutional Review Board – please clarify it.
  • I have serious doubts regarding this section because many details are missing.
    • Study sample – how were the study participants chosen to participate in the study, what was the procedure of sampling? Why were they selected only from 2 universities? Why were the first and second year students excluded from the study?
    • Applied questionnaire – did Authors you a validated questionnaire? I suppose that within this questionnaire, study participants had to answer each question in regard to pre-pandemic and pandemic period, however this generates a risk of bias. This should be clarified.

Discussion

  • Similarly as in the Introduction, Authors should try to cite the most topical studies which were carried out in the COVID period.
  • Authors must add limitations of this study (for example the risk of bias in terms of applied questionnaire which I indicated earlier and small number of study participants)

Author Response

Thank you very much for your pertinent and valuable recommendations.

Considering the suggestions received from the reviewers, the following changes were made to the article.

Corrections were made to the Abstract section, as well as to the Introduction section.

Subsequent paragraphs of the Material and Methods section were numbered.

Students from only 2 universities were chosen because that's how we succeeded in communication. Initially, the questionnaire was sent to a higher number of students.

 The first- and second-year students were eliminated considering that they did not go through the preparation module regarding infection control, which could have led to processing errors.

To avoid other misunderstandings, we rewrote the protocol, taking into account the 126 students.

The questionnaire was evaluated by a panel of experts from the Faculty of Dentistry, following a qualitative pre-testing of the content, followed by its validation. The questionnaire was openly applied. The questionnaire was uploaded online on the Google docs platform.

 All the information obtained was collected after the pandemic, with the establishment of the level of knowledge at that level.

I improved the clarity of the tables.

I gave up some acronyms.

I corrected the sentences that started with a percentage.

I renamed the subjects as students.

I made the corrections for Discussions.

I removed attitudes.

I am attaching the corrected form of the article.

Round 2

Reviewer 2 Report

Authors have improved the manuscript, although I still have some concerns with the Discussion section. "In a study 338 conducted in Nigeria, the majority of respondents (83%) had high knowledge of HH, due 339 to the development of student training programs [27]. On the other hand, Van de Mortel 340 et al. observed that only 63% of medical students included in their study knew the 5 341 moments of HH [28]." - it is still unknown whether these cited studies were conducted in a period of COVID or not.

There are no limitations of the study which I mentioned previously,  for example the risk of bias in terms of applied questionnaire and small number of study participants 

Author Response

Following your requests, we have made the requested changes, regarding the period of carrying out the cited studies 27,28, being made in the period prior to the pandemic.

We mentioned the limitations of this study and the risk of bias.

I also obtained the opinion of the ethics commission of the university for the applied questions and I mentioned in the paper.

I corrected the English language with the help of a native English speaker.

I corrected the way of writing in the bibliography.

Thank you very much for the relevance of the approaches.

With special respect and consideration,

Iulia Saveanu